# Supplementation of 17β-Estradiol Normalizes Rapid Gastric Emptying by Restoring Impaired Nrf2 and nNOS Function in Obesity-Induced Diabetic Ovariectomized Mice

**DOI:** 10.3390/antiox9070582

**Published:** 2020-07-03

**Authors:** Jeremy C. Sprouse, Chethan Sampath, Pandu R. Gangula

**Affiliations:** 1School of Graduate Studies, Meharry Medical College, Nashville, TN 37208, USA; jsprouse12@email.mmc.edu; 2Department of ODS & Research, School of Dentistry, Nashville, TN 37208, USA; csampath@mmc.edu

**Keywords:** gastroparesis, estrogen, antioxidants, cytokines, progesterone, diabetes, obesity, Nrf2, nNOS

## Abstract

Gastroparesis (Gp) is a multifactorial condition commonly observed in females and is characterized by delayed or rapid gastric emptying (GE). The role of ovarian hormones on GE in the pathogenesis of obesity induced type 2 diabetes mellitus (T2DM) is completely unknown. The aims of our study are to investigate whether supplementation of 17β-estradiol (E_2_) or progesterone (P_4_) restores impaired nuclear factor erythroid 2-related factor 2 (Nrf2, an oxidative stress-responsive transcription factor) and nitric oxide (NO)-mediated gastric motility in ovariectomized (OVX) mice consuming a high-fat diet (HFD, a model of T2DM). Groups of OVX+HFD mice were administered daily subcutaneous doses of either E_2_ or P_4_ for 12 weeks. The effects of E_2_ and P_4_ on body weight, metabolic homeostasis, solid GE, gastric antrum NO-mediated relaxation, total nitrite levels, neuronal nitric oxide synthase (nNOSα), and its cofactor expression levels were assessed in OVX+HFD mice. HFD exacerbated hyperglycemia and insulinemia while accelerating GE (*p* < 0.05) in OVX mice. Exogenous E_2_, but not P_4_, attenuated rapid gastric emptying and restored gastric nitrergic relaxation, total nitrite levels, nNOSα, and cofactor expression via normalizing Nrf2-Phase II enzymes, inflammatory response, and mitogen-activated protein kinase (MAPK) protein expression in OVX+HFD mice. We conclude that E_2_ is beneficial in normalizing metabolic homeostasis and gastric emptying in obese, diabetic OVX mice consuming a fat-rich diet.

## 1. Introduction

Gastroparesis (Gp) is a chronic disease that presents with various clinical symptoms including early satiety, nausea, vomiting, and mild to severe abdominal pain. Gp and Gp-like syndromes are hallmarked by an objective finding of abnormally delayed and/or accelerated gastric emptying (GE) in the absence of gastric outlet obstruction [1,2,3]. The most common etiologies of Gp include idiopathic, post-surgical/iatrogenic, and a severe complication of uncontrolled diabetes mellitus (DM) [4]. Numerous lines of evidence suggest the loss of enteric neuronal nitric oxide synthase (nNOS) and nitric oxide (NO)-mediated relaxation of gastric smooth muscle as the primary culprits of Gp in diabetic patients [5,6,7,8]. In addition, oxidative stress associated with hyperglycemia and obesity are known contributors to the pathogenesis of diabetic Gp [9,10]. Furthermore, approximately 20% of diabetic patients exhibit accelerated GE while 30–40% show delayed GE. Growing evidence suggests rapid or accelerated GE to be indicative of early type 2 diabetes mellitus (T2DM) induction and hyperinsulinemia, while others suggest these insults may influence mitogen-activated protein kinase (MAPK) signaling and interstitial cells of Cajal (ICC) architecture [11]. Although much of the research in Gp has been devoted to delayed GE, increased efforts to uncover the mechanisms involved in rapid GE are needed to understand the complexities of Gp or Gp-like symptoms.

Although women comprise the vast majority of Gp patients presenting with abnormal GE, the influence (presence or absence) of ovarian hormones on GP remains unclear in human and animal studies. Young adult healthy women and female rodents exhibit slower basal gastric motility than men due to increased circulatory estrogens and neuronal nitric oxide synthase (nNOS)-NO activity in gastric smooth muscle [12,13]. Similarly, some studies observe slower GE in females during the luteal phase of the menstrual cycle when estrogen and progesterone levels are elevated [14,15]. Sex hormones, particularly 17β-estradiol (E_2_) and progesterone (P_4_), also play an important role in modulating GE in pre-and post-menopausal women [13,16]. Postmenopausal women without hormone replacement had faster rates of solid GE similar to those of men, suggesting a critical role of sex hormones in slowing GE. Menopausal women treated with sex hormone replacement therapy had a decreased rate of gastric emptying of solids compared with men. The above studies suggest that, in general, gastric emptying is slower in women than age matched men due to elevated levels of sex hormones.

Female sex steroid hormones, E_2_ and P_4,_ are involved in non-genomic and genomic transcriptional activity mediated by their respective receptors, ERs and PRs [17]. While also playing a vital role in numerous physiological and pathological states in both men and women, ER and PRs influence several cellular targets [18]. Importantly, E_2_, P_4,_ and nNOS exhibit interactions in several tissue types [19,20,21]. E_2_ has been suggested to improve nNOS expression in gastric neuronal cells and neutrophils through either genomic or non-genomic actions, mediated by specialized ER subtypes [22]. Within the gastrointestinal (GI) system, E_2_ is thought to influence gastric parietal cell secretion in association with nNOS by diffusing to and acting on adjacent inhibitory nNOS neurons to regulate gastric emptying [23]. Uncovering the mechanisms, differentiating motor abnormalities in Gp are illusive due to the complexities in animal models and ovarian hormones status. Though many human and animal Gp studies focus on the effects of E_2_, several studies have reported diverging effects of P_4_ on stomach motility [24]. PR signaling is proposed to regulate G-protein expression levels in female chronic constipation studies [25]. Though the mechanisms responsible for the non-genomic effects of P_4_ are not fully understood, it has been proposed that P_4_, through PR, may lead to rapid activation of tyrosine kinases and phospholipases, mitogen-activated protein kinase, or inhibition of membrane transport systems [26,27]. Furthermore, the types of changes induced by the non-genomic actions may be tissue specific, as diverse effects have been demonstrated in muscle, neural, endocrine, and reproductive cells. In this study, we sought to identify gastric mechanisms of ER and PR in regulating motility in obese, diabetic female rodents.

Findings from human and rodent models of obesity and DM suggest that oxidative stress and the subsequent inflammation are important factors contributing to many pathologic conditions, including gastrointestinal dysmotility [28]. Attention to oxidative stress regulators such as Nrf2, a stress-responsive transcription factor, has developed due to the potential effects in alleviating GI complications of DM, specifically diabetic Gp [29]. Although estrogens are widely recognized to exert antioxidant and regulatory functions on the immune system and the expression of cytokines, whether these hormones alter the inflammatory response or the antioxidant regulatory enzymes (Nrf2 and phase II enzymes) observed in animal models of insulin resistance and type 2 diabetes has not been addressed to date. Furthermore, it has not been established whether this pathway could represent an effective therapeutic target to fight against gastrointestinal disturbances induced by a high-fat diet (HFD) and ovariectomy (OVX). Most importantly, conflicting studies exist showing that endogenous E_2_ and P_4_ are implicated in predisposing women to experiencing Gp and Gp-like syndromes [30,31,32].

Although management of diabetic Gp can be achieved, current treatments require prokinetic drugs with limited efficacy or expensive surgically invasive procedures or mandates extensive dietary measures in metabolically compromised T2DM patients to reduce symptoms. Rodent studies using clinically relevant (obese, diabetic) models are critical to understanding how predisposing factors such as ovarian hormone status influence the development of Gp. Therefore, the aim of this study was to determine whether E_2_ and P_4_ treatment influences gastric nNOS/Nrf2 and cellular protein targets in addition to NO-mediated gastric motility and GE after chronic ovariectomy and obesity-induced T2DM in mice.

## 2. Materials and Methods

### 2.1. Experimental Design

All experiments were approved by the Institutional Animal Care and Use Committee (IACUC) at Meharry Medical College (MMC), in accordance with recommendations of the National Institutes of Health (NIH) Guide for the Care and Use of Laboratory Animals. Adult female C57BL/6J (12–15 weeks old) mice were purchased from Jackson Laboratories (The Jackson Laboratory, Bar Harbor, ME, USA), the approved protocol ID: #17-09-764. All animals were housed in the institutional animal care vivarium under standard conditions (4 mice/cage, 12 h light cycle) and allowed access to food and water ad libitum.

### 2.2. Surgical Ovariectomy, HFD Feeding, and 17β-Estradiol and Progesterone Replacement

Initially, mice were grouped into sham and OVX surgery groups. All procedures were performed using aseptic technique. Ovariectomy was performed as reported earlier [14,33]. Under gas anesthesia (2% isoflurane), the ovaries were exposed via bilateral flank incisions and precisely excised. The incision was approximated with stainless steel wound clips to achieve primary healing. All mice were allowed to recover for 10 days prior to regrouping.

After a 10-day resting period, each group (Sham and OVX) was randomized into two new groups based on the type of diet offered: mice fed the standard chow (ND), Sham +ND (*n* = 6), and OVX+ND (*n* = 6) groups and mice fed high-fat diet (HFD), Sham+HFD (*n* = 6), and OVX+HFD (*n* = 30) groups. Once feeding began, fresh chow was provided daily, and any remaining chow from the previous day was discarded. In addition to HFD feeding, groups of OVX+HFD mice were further grouped to receive daily subcutaneous doses of either 17 β- estradiol (E_2_; 0.25 mg/kg bw (*n* = 6), or 1.0 mg/kg bw (*n* = 6)) or progesterone (P_4_; 1.0 mg/kg bw (*n* = 6), or 4.0 mg/kg bw (*n* = 6)) throughout the study. These doses of sex steroid hormones were chosen based on previous reports of circulating estradiol levels that are in the peak pharmacological and supra-physiological [34,35,36,37] ranges. Body weights and blood glucose levels were measured weekly to monitor obesity and diabetes induction. At the end of the study, gastric emptying and non-adrenergic non-cholinergic relaxation (NANC) relaxation were measured, and tissues were snap frozen and stored in −80 °C.

### 2.3. Intraperitoneal Glucose Tolerance Test

To assess glucose tolerance and induction of T2DM, the intraperitoneal (i.p) glucose tolerance test (IPGTT) was performed on fasted mice during the 11th week [38]. Groups of mice were fasted for 6 h with water ad libitum. Blood glucose concentrations were analyzed prior to an intraperitoneal glucose load (2 g/kg body wt ip) and subsequently at 30, 60, 90, and 120 min after. Blood glucose levels were measured via tail vein sample using a standard glucometer. The values were plotted, and area under curve (AUC) for each group was calculated.

### 2.4. Measurement of 2 h Solid Gastric Emptying

Gastric emptying experiments for all groups of mice were performed as published [38,39,40]. At the end of the 12 week treatment period, mice were fasted overnight. At the start of the protocol, each mouse was singly housed and provided a pre-measured bolus of food (ND or HFD) with water ad libitum for three hours. Afterwards, the mice were moved to clean cages and fasted for an additional two hours; the remaining food was dried and weighed to determine the food intake (FI). To measure the rate of gastric emptying, mice were euthanized by cervical dislocation, and stomachs were carefully dissected by the same experimenter to minimize variation. Full and empty stomach weights were recorded; the difference estimated the remaining gastric content (GC) after 2 h of fasting. The rate of GE was measured with the equation of 1- [(GC/FI) × 100].

### 2.5. Electric Field Stimulation to Elicit Nitrergic Relaxation in Gastric Antrum Neuromuscular Specimens

Electric field stimulation (EFS)-induced non-adrenergic non-cholinergic relaxation (NANC) was examined in circular gastric antrum neuromuscular strips in WT mice (*n* = 4/group) as previously described [38]. The circular gastric antrum neuromuscular strips were mounted in 10 mL Krebs buffer at 37 °C, and NANC-dependent nitrergic relaxation (nNOS function) was determined at 2 Hz 26 (DMT Technologies, Nottingham, UK). The NO dependence of nitrergic relaxation was confirmed with NG-nitro-L-arginine-methyl ester treatment (L-NAME, 100 μM, 30 min). Comparison between groups was performed by measuring the area under the curve (AUC/mg of tissue) of the EFS-induced relaxation (AUCR) curve at 1 min and the baseline (AUCB) curve at 1 min, as follows: (AUCR–AUCB)/weight of tissue (mg) = AUC/mg of tissue.

### 2.6. Estimation of Serum 17β-Estradiol, Progesterone, Insulin, and Total Nitrite Concentration

After euthanization, blood was collected via cardiac puncture. Serum was isolated and stored at −80 °C until analyzed. Serum levels of 17β- estradiol (BioVision, Inc., Milpitas, CA, USA), progesterone (Crystal Chem, Elk Grove Village, IL, USA) and insulin (Crystal Chem, Elk Grove Village, IL, USA) were measured using ELISA kits per manufacturers’ guidelines. Serum total nitrite was measured via colorimetric assay based on manufacturer’s protocol (BioVision, Inc., Milpitas, CA, USA) [41]. As in previous reports, the homeostatic model of insulin resistance (HOMA-IR) index was calculated as (fasting serum glucose × fasting serum insulin/22.5) to assess insulin resistance [33].

### 2.7. Gel Electrophoresis and Western Blot Analysis

Snap-frozen gastric antrum specimens were homogenized with sonication in radioimmunoprecipitation assay (RIPA) buffer containing a protease inhibitor (Thermo Scientific, Rockford, IL, USA). Protein concentrations were estimated in each lysate via bicinchoninic acid (BCA) method. Equal concentrations of lysates (40 µg) were separated on 6% and 12% SDS polyacrylamide gels prior to wet transfer to a nitrocellulose membrane. Each membrane was subsequently blocked with 5% dried non-fat milk for 1 h, then incubated with primary polyclonal antibody (GCH-1, (1:500), DHFR (1:500), IL-6 (1:500), TNF-α (1:500), Nrf2 (1:1000) purchased from (Santa Cruz Biotechnology, Santa Cruz, Ca, USA)) overnight (~18 h), respectively. Expressions of nNOSα (N-terminal) and MAPK were detected using rabbit polyclonal antibodies at 1:1000 (Abcam, (Cambridge, MA, USA) and Cell Signaling, Inc (Danvers, MA, USA), respectively). The membranes were washed 3 times for 10 min each in 0.01% TBS-Tween, then incubated in horseradish peroxidase-conjugated secondary antibody (1:1000) for 1 h at room temperature. The blots were visualized with ECL Western Blotting Detection Reagent (GE Healthcare Bio-Sciences Corp., Piscataway, NJ, USA), and the reactive bands were analyzed quantitatively by optical densitometry. The blots were stripped and re-probed to measure protein expression. Blots were re-probed with β-actin polyclonal antibodies (1:5000) (Abcam, Cambridge, MA, USA) to enable normalization of signals between samples.

### 2.8. Statistical Analysis

Data were presented as the mean ± standard error of the mean (SEM). Statistical comparisons between groups were determined by the Tukey’s test after one-way Analysis of Variance (ANOVA) using the GraphPad Prism Version 5.0 (GraphPad Software, San Diego, CA, USA). A *p*-value of less than 0.05 was considered statistically significant.

## 3. Results

### 3.1. Supplementation of E_2_ and P_4_ Prevent Body Weight Gain, Adiposity and Hyperglycemia in Obese, Diabetic OVX Mice

Weekly blood glucose and body weights were measured to confirm the onset of obesity and obesity-induced DM in intact and ovariectomized mice while consuming an HFD over 12 weeks. In addition, visceral adipose tissue weight and serum insulin concentration were also measured. As shown in Table 1, we observed HFD-fed mice were more susceptible to body and adipose weight gain when compared to ND-fed groups (Sham+HFD: 19.25 ± 2.61 vs. Sham+ND: 3.39 ± 1.01) (*p* < 0.05) (OVX+ND: 4.39 ± 1.25 vs. OVX-+HFD: 13.82 ± 0.68). HFD-fed mice gained greater than three times more weight than ND-fed counterparts. Interestingly, no differences in weight gained were noticed over the 12-week between Sham+ND or OVX+ND mice. OVX mice consuming ND demonstrated increased fasting glucose and insulin levels, suggesting a protective role of ovarian hormones in regulating metabolic homeostasis. Moreover, mice (Sham and OVX) consuming HFD over the 12 week period display increased body weights, visceral adipose tissue weights, and circulatory concentrations of glucose and insulin, consistent with obesity-induced DM. Consistent with our hypothesis and previous reports, E_2_ and P_4_ supplementation markedly decreased fasting glucose levels, visceral adipose weight, and weight gain in HFD-fed OVX mice [18,42,43]. Fasting insulin levels were unchanged in mice receiving E_2_ and P_4_, as shown in Table 1.

### 3.2. E_2_ and P_4_ Prevent Insulin Resistance and Glucose Intolerance in HFD-fed OVX Mice

To determine the ability of E_2_ and P_4_ supplementation to prevent HFD-induced disturbances in insulin sensitivity and glucose tolerance, we calculated the homeostatic model of insulin resistance (HOMA-IR) from fasted insulin concentration and performed intraperitoneal GTT in OVX mice. The intraperitoneal glucose tolerance test (IPGTT) is used in clinical practice and rodent studies to identify impaired glucose tolerance in patients with type 2 diabetes [44]. Here, we present the data in two ways: the linear plot of the two-hour IPGTT in Figure 1A and the AUC_IPGTT_ in Figure 1B. AUC_IPGTT_ was significantly (*p* < 0.05) elevated in mice exposed to the HFD. Furthermore, E_2_ and P_4_ prevented fasting hyperglycemia and glucose tolerance in HFD-fed OVX mice. Interestingly, in ND-fed mice, ovariectomy did not influence fasting glucose level during the glucose tolerance tests. When HOMA-IR indices were calculated, there were significant increases with OVX and/or HFD-feeding. The HOMA-IR was markedly higher in HFD+OVX mice than in any other treatment group (Sham+ND, ND+OVX and Sham+HFD) (Figure 1C). E_2_ and P_4_ supplementation reversed elevated HOMA-IR in HFD-fed OVX mice. Furthermore, our proof-of-principle findings are consistent with previously reported data in HFD-fed OVX mice [18].

### 3.3. Serum Estradiol, Progesterone, Testosterone, Adiponectin, and Total Nitrite Concentration

Serum concentrations of sex steroid hormones [17 β-estradiol, E_2_, progesterone, P_4_], adiponectin, and total nitrite were measured using commercially available kits and are presented in Table 2. Serum total nitrite is a representative measurement of nitric oxide production in systemic circulation. When compared to levels in Sham+ND, total nitrite was decreased in Sham+HFD and OVX+ND groups but maintained in HFD-fed OVX mice. E_2_ supplementation further elevated total nitrite concentration in the serum of OVX+HFD mice at the lower concentration. Moreover, E_2_ and P_4_ concentrations were elevated in serum samples of HFD-fed (Sham and OVX) mice to levels comparable to Sham+ND control mice. Both E_2_ and P_4_ were decreased in chronic OVX+ND when compared to the Sham+ND group. Serum progesterone levels were repressed in E_2_-treated mice and maintained in P_4_-treated mice. Similarly, adiponectin is an adipokine that is involved in glucose and lipid metabolism that is exclusively secreted from differentiated adipocytes (4). Circulating levels of adiponectin were measured in female mice and markedly decreased in the serum of Sham+HFD, OVX+ND and OVX+HFD groups when compared to Sham+ND. Moreover, E_2_ and P_4_ rescued elevated serum adiponectin levels in HFD-fed OVX mice. Although circulating E_2_ and P_4_ concentrations have been characterized in previous reports, the systemic concentrations of total nitrite and adiponectin levels have not been examined in these conditions.

### 3.4. Supplementation of E_2_ Restored Impaired Gastric Emptying and Nitrergic Relaxation

As shown in Figure 2A, we observed a significantly (*p* < 0.05) delayed (55% of ND control) gastric emptying in Sham+HFD mice when compared to Sham+ND (72% (ND) vs. 38% (HFD) group. Chronic loss of ovarian hormones (12 weeks after OVX) in female mice also yielded a similar (44%) delay in gastric emptying (Sham+ND 72% vs. OVX-ND 41%). Interestingly, solid GE was significantly (*p* < 0.05) faster in HFD-fed OVX mice when compared to ND-fed Sham and OVX groups. When supplemented with doses of E_2_ (0.25 mg/kg) and P_4_ (4mg/kg), gastric emptying was normalized to nearly 70% of the rapid GE rates observed in OVX+HFD mice (*p* < 0.05). No changes in GE were noticed with other doses of both hormones in OVX+HFD group of mice.

Similarly, low frequency electric field stimulation (2 Hz) of gastric antrum neuromuscular specimens elicits NANC relaxation in an organ bath setup. NO-mediated nitrergic relaxation comprises the primary component of NANC relaxation in gastric smooth muscle. NANC relaxation was severely impaired in the diabetic (Sham+HFD) and chronic OVX (+ND) state, consistent with delayed GE rates. Although GE was accelerated in HFD-fed OVX mice, we also observed significantly impaired NANC relaxation further than in ND-fed OVX mice (Figure 2B). Only the low dose of E_2_ (0.25 mg/kg) restored NANC relaxation in gastric antrum of HFD-fed OVX mice (*p* < 0.05). Blockade of nNOS activity with L-NAME significantly abolished relaxation to confirm this relaxation was mediated by NO (*p* < 0.05). Taken together, E_2_ restored nitrergic function to normalize GE rates in OVX+HFD mice.

### 3.5. nNOSα Expression Is Maintained in OVX+HFD Mice, but BH_4_ Cofactor Biosynthesis Is Impaired in Gastric Antrum

After observing functional changes in nitrergic relaxation in the gastric antrum, we next measured whether nNOSα expression was altered in obesity-induced diabetes via western blotting methods. Our results show that the nNOSα expression level in Sham+HFD and OVX+ND was significantly (*p* < 0.05) lower than that of control mice (Sham+ND) (Figure 3A). Surprisingly, a similar change in nNOSα expression was noticed in OVX+HFD mice (Figure 3A). Interestingly, the protein expressions of nNOSα were significantly elevated in E_2_-treated mice (0.25 mg/kg), though they were unchanged in the other treatment groups.

Sepiapterin (BH_4_) is an essential cofactor for nNOS coupling and activity. The biosynthesis of BH_4_ (cofactor for nNOS) is chiefly regulated by two enzymes, GCH-1 (de novo) and DHFR (salvage). Although we observed no change in GCH-1 expression in HFD-fed Sham-operated mice, GCH-1 protein expression was elevated in OVX+ND mice and was significantly (*p* < 0.05) repressed in HFD-fed OVX mice (Figure 3B). Both E_2_ and P_4_ significantly (*p* < 0.05) increased GCH-1 expression in OVX+HFD mice. Interestingly, we found DHFR expression was significantly upregulated in instances where GCH-1 expression was diminished in HFD-fed groups (*p* < 0.05) (Figure 3C). In addition, E_2_ and P_4_ had dose-differential effects on GCH-1 and DHFR expression, suggesting hormonal regulation of BH_4_ synthesis enzymes.

### 3.6. E_2_ and P_4_ Supplementation Regulates MAPK, ER and PR Levels in OVX+HFD Mice

Activated MAPK expression levels were examined, as hyperglycemia and estradiol are known to influence the activation of MAPK and intracellular pathways [17,45]. Here, we observed a significant (*p* < 0.05) decrease in MAPK expression with chronic OVX but not HFD in Sham mice. Similarly, MAPK was upregulated in HFD-fed OVX mice (Figure 4A). E_2_ and P_4_ treatment exacerbated the upregulation in OVX+HFD groups. Moreover, representative western blots of ERα and ERβ levels in OVX mice following HFD feeding are presented in Figure 4B,C. Our data indicate that HFD feeding is associated with a depletion in sex hormone receptors as compared to ND-fed mice. Furthermore, OVX+ND mice displayed a decreased expression of ER that was significantly depleted in OVX+HFD groups. Sex hormones, both E_2_ and P_4_, improved ER and PR expression (Figure 4B–D).

### 3.7. Gastric Pro-inflammatory Cytokine (IL-6 and TNFα) are Elevated in HFD-fed Mice and Restored by E_2_ and P_4_ Supplementation

Emerging research suggest that inflammatory cytokines are known to influence gut function [46,47]. Gastric antrum homogenates from the mice were examined for expression of TNF-α and IL-6 by western blotting (Figure 5A,B). The analysis results revealed that the expression levels of IL-6 did not differ significantly with OVX, whereas the expression levels of TNF-α were much higher than in the control groups (*p* < 0.05). Moreover, HFD consumption led to a significant increase in the relative quantity of TNF-α and IL-6 in comparison to ND-fed mice. HFD-fed mice treated with E_2_ displayed diminished pro-inflammatory tissue expression of TNF-α and IL-6 (*p* < 0.05).

### 3.8. E_2_ and P_4_ Regulate Nrf2 and Phase II Antioxidant Enzymes Expression in OVX+HFD Mice

As reported, obesity and hyperglycemia are associated with increased oxidative stress [48,49]. Although ovarian hormones demonstrate antioxidant properties, Nrf2 and its downstream Phase II enzyme have emerged as key players in combating cellular oxidative stress insult [29,50]. Induction of Nrf2 expression was measured via western blotting (Figure 6A). Nrf2 protein expression was significantly ablated in HFD-fed mice, consistent with our previous findings in obese diabetic mice [38]. Nrf2 expression was maintained in chronic OVX (OVX+ND). Furthermore, estradiol repressed (*p* < 0.05) Nrf2 induction in gastric antrum to a greater extent than progesterone in HFD-fed OVX mice; P_4_-treated mice retained Nrf2 expression. Similar observations were seen with Phase II enzyme induction (Figure 6B–D). Catalase, superoxide dismutase (SOD), and heme oxygenase 1 (HO-1) were maintained in HFD-fed Sham mice compared to ND-fed counterparts (Figure 6B–D). OVX led to an upregulation of SOD1, although decreased expression of cellular Catalase and HO-1 was observed (Figure 6B–D). E_2_ and P_4_ administration differentially regulated Phase II antioxidant enzyme expression; both increased expression of SOD, HO-1, and Catalase at different concentrations (Figure 6B–D).

## 4. Discussion

Gp is commonly observed as a sequela of obesity induced (T2DM) diabetes. It is a condition characterized by chronic stomach dysmotility leading to abnormal GE of solids and/or liquids in the absence of mechanical obstruction. Although the influences of hyperglycemia, oxidative stress, and inflammation have been implicated in the development of Gp in obese or diabetic state, the role of sex steroid hormones in GI motility is elusive. It is well known that younger women in particular are more susceptible for gastroparesis than age-matched men [12]. In addition, GI motility rates have been suggested to fluctuate during menstruation, pregnancy, and with menopause [16]. In this study, we assessed the combined effects of obesity and the loss of hormone (chronic ovariectomy) on metabolic and gastric parameters in female mice. Our major experimental aim was to determine whether sex steroid hormones were effective in restoring abnormal gastric emptying and nitrergic relaxation in obese, diabetic OVX mice consuming a fat-rich diet. The findings are that HFD feeding in SHAM-operated and OVX mice leads to significant metabolic changes, including increased body mass, adiposity, glycemia, insulin levels, and pro-inflammatory cytokine markers. Furthermore, we demonstrate delayed GE in HFD-fed ovary-intact mice, whereas OVX mice consuming an HFD exhibit accelerated GE and impaired nitrergic relaxation, mimicking the characteristics observed in T2DM and post-menopausal women [7,51]. While E_2_ and P_4_ supplementation was beneficial in normalizing metabolic factors commonly observed in obesity-induced diabetes, only E_2_ contributed to attenuating delayed GE by restoring nNOS-mediated nitrergic relaxation in OVX+HFD mice. These findings contribute to Gp field by unraveling the complex nature of ovarian hormones and their roles in obesity, T2DM, and GI motility (Figure 7).

Ovarian hormones are known to participate in energy homeostasis, adipose distribution, and glucose and insulin sensitivity [52]. Estrogen in particular is known to reduce the incidence of type 2 diabetes in postmenopausal women, though the mechanism is not well understood. Our studies demonstrate that the combination of OVX and HFD increased body weights, glycemia, and insulinemia and led to significantly increased GTT_AUC_ and HOMA-IR. These data are in agreement with previous reports of hyperglycemia, hyperinsulinemia, and insulin resistance observed in postmenopausal women, obese patients, and rodent models [33]. Furthermore, these data corroborate several reports suggesting that OVX increases susceptibility to weight gain and fat mass increase when fed with HFD, previously reported to be secondary to high degrees of adipose tissue inflammation and pro-inflammatory cytokines production [42,53]. Our studies further demonstrate that E_2_ and P_4_ were beneficial in restoring metabolic homeostasis in HFD-fed OVX mice including insulinemia, glycemia, glucose tolerance, and HOMA-IR.

Additionally, in our experimental models, we characterized the systemic concentrations of E_2_ and P_4_, total nitrite, and adiponectin. HFD consumption in OVX mice was consistent with elevated E_2_ and P_4_ levels. We recently demonstrated that prolonged HFD consumption elevated serum E_2_ and obesity markers [38,54]. Though not characterized in this study, it is important to note that the biosynthesis of estrogens differs between premenopausal and postmenopausal women and rodent models (53). Premenopausal women mainly synthesize estrogens in the ovary, whereas estrogen biosynthesis is replaced by peripheral site synthesis (mainly by aromatases in adipose tissue) in postmenopausal women and rodent models [55,56]. Similarly, we assert that systemic concentrations of adiponectin are elevated with increased adiposity and may be normalized by E_2_ and P_4_ treatment. D’Eon et al. also demonstrated that E_2_ decreased adiponectin levels in ovariectomized mice in association with a reduction in adiposity [57]. Conflicting and very few reports exist in the literature on the exact function of adiponectin in GI; this topic deserves more attention as a regulator of obesity and inflammation. Moreover, it is important to note that the variations in methodologies reported in the literature may influence the actual levels of hormones and/or their gastric receptors localized in enteric neurons, often resulting in different interpretations of the effects of sex hormones on regulating gastric motility and emptying.

Normal gastric emptying results from an integration of excitatory (cholinergic and adrenergic) contractions and inhibitory (primarily NO) relaxation of the gastric fundus, antrum, pylorus, and proximal duodenum. The enteric nervous system (ENS) regulates the GI tract through complex interactions between smooth muscle, enteric and autonomic nerves, and the interstitial cells of Cajal (ICC), the pacemaker cells of the gut. The release of nitric oxide (NO) by neuronal NO synthase (nNOS) from non-adrenergic, non-cholinergic (NANC) neurons has been shown to be an important factor controlling gastrointestinal motility and transit time [8]. Interestingly, diabetic and idiopathic Gp patients lack functional coordination between these cell types and are suggested to be disrupted. In particular, accelerated gastric emptying is often observed in early diabetic patients and rodent models of diabetes and genetic knockdown models (Ghrelin, Leptin null models) [11]. Commonalities between these and our HFD-fed OVX model include obesity, hyperglycemia, hyperinsulinemia, inflammation, and oxidative stress. Here, for the first time, we demonstrated that rapid GE in HFD-fed chronic OVX mice is associated with impaired nitrergic input. This finding opposed the delayed GE observed in ND-fed OVX mice or in HFD-fed, ovary-intact mice.

Our data also suggest that sex steroids may play an important role in modulating the nNOS/NO system in the stomach, similar to others shown in the literature [58,59,60]. OVX mice with a chronic depletion of sex hormones display delayed GE and reduced nitrergic relaxation when compared to control (SHAM+ND), suggesting that, in addition to enteric neurons, the smooth muscle or ICC milieu may be affected. Interestingly, HFD-fed OVX mice displayed significantly faster GE rates. The daily consumption of an HFD increased inflammatory stress caused by obesity and HG. We speculate that the sex hormone depleted (OVX) gastric milieu may exhibit compensatory mechanisms to facilitate faster GI motility when consuming a daily HFD in the gastric tissue. Perhaps a GI system devoid of ovarian sex hormones (chronic OVX) would display gradual changes in ICC, smooth muscle, and enteric neurons distribution to possibly “rewire” the female gut. Furthermore, supplementation of E_2_ and P_4_ restored not only systemic and gastric fitness through restoration of proinflammatory pathways but also nNOS and its essential BH_4_ cofactor expression to normalize GE rates in OVX+HFD mice comparably to those observed in ND-fed counterparts. In addition, only E_2_-treated mice displayed increased gastric NANC and serum total nitrite levels to normalize accelerated GE. Many reports have demonstrated that estrogen supplementation improved nNOS expression, nNOS neurons, and, finally, nNOS function in adult OVX but not in HFD fed rodents [19,60]. Shah et al. demonstrated the increases in nNOS protein were mainly localized to the neurons in the myenteric plexus and the nerve axons coursing in parallel with the inner circular muscle [58]. In the current study, we demonstrated that E_2_ but not P_4_ attenuated (1) diminished nNOS protein expression, (2) nitrergic relaxation (as confirmed by using NOS inhibitor L-NAME), and (3) total nitrite production, which is correlated with restoring GE in OVX+HFD mice. Future studies will be directed to investigate the cellular changes in nNOS, ERs, and Nrf2 in female gastric specimens at various time points among these groups. Furthermore, since there is no estrogen-responsive element in the promoter region for nNOS gene, the mechanism by which E_2_ regulates nNOS gene expression remains elusive, and this is the subject for future studies. In addition, our results are in line with in-vitro studies that demonstrate estrogen and progesterone effects on contractile response and myoelectric activity of the gastrointestinal smooth muscle [31]. Although our studies focus on the role of E_2_ and P_4_ in modulating the nitrergic component of the NANC nerves, other inhibitory mediators such as vasoactive intestinal polypeptide (VIP) from the NANC nerves may also be involved, though not explored.

Sex hormones have been shown to facilitate cellular changes through myriad mechanisms. Recent studies have implicated estradiol in regulating energy metabolism, inflammation, and antioxidant properties [61,62]. ER signaling mechanisms are diverse and include membrane-bound estrogen receptors to facilitate many intracellular actions within the various cell types. Particularly, the non-genomic activity of ERs is mediated by both cytosolic (ERα,β) and membrane-bound (G-protein receptor 30 (GRP30/GPER) and post-translationally modified membrane bound ERs (m-ERα/m-ERβ) [17,63]. Although GPR30 is known to regulate many of the non-genomic signaling cascades, including MAPK and Protein Kinase B (AKT), that occur within minutes to hours of activation, cytosolic ERs are widely known to interact with MAPK [64]. Since this study primarily focused on the effects of E_2_ and P_4_ in chronic OVX+HFD (12 week model) animals, our aim was to elucidate the genomic changes as opposed to the quick, often transient non-genomic actions. ERs have distinct tissue expression patterns in both humans and rodents. In the “classical” mechanism of estrogen action, estrogens diffuse into the cell and bind to cytosolic ERs, then translocate into the nucleus to regulate gene transcription through interactions with the consensus estrogen response element (ERE) sequence [17]. In our studies, we hypothesized that E_2_ could regulate expression of other cellular mediators through genomic effects driven by both ERs. Here, we demonstrated that ERα and ERβ expressions were enhanced with E_2_ and P_4_ treatment and may mediate the genomic effects. Most importantly, estrogens are known to protect various cell types (including neuronal cells) against oxidative stress; however, the integral role of these hormones associated with oxidative stress-induced gastroparesis is yet to be demonstrated [4,15].

In the same regard, P_4_ is a steroid hormone that has been suggested to regulate smooth muscle contraction in various regions in the GI tract through PR. Much attention is needed to understand the effects of P4 and PR in Gp. Interestingly, P_4_ has been observed to downregulate G-coupled protein receptors (GPCRs), specifically those that mediate contraction and relaxation of gastric smooth muscle cells via the Rho kinase pathway [27]. Al-Shboul further proposed that hormonal effect on muscle contraction represents mostly a non-genomic action of PR in rat gastric smooth muscle. PR has been localized in stomach and colon muscle cells [25]. Similar to the effect of progesterone, estrogen was also reported to reduce the contraction of female GSMCs through activation of the NO pathway [27,65]. In this study, we demonstrate that E_2_ and P_4_ supplementation influences gastric function through their receptors and maintains normal gastric emptying in HFD induced diabetic rodents.

The role and the regulation of nNOS by sex hormones, but most importantly by its cofactor, BH_4_, are well documented [4]. It is commonly accepted that GCH-1 mediates de novo BH_4_ synthesis, while DHFR is involved in the salvage pathway production for the nNOS cofactor. In our study, both E_2_ and P_4_ significantly increased GCH-1 expression in OVX+HFD mice. Moreover, E_2_ and P_4_ had dose-differential effects on GCH-1 and DHFR expression, suggesting hormonal regulation of BH_4_ synthesis enzymes. Importantly, ovarian hormone supplementation is known to be either beneficial or detrimental depending upon the doses tested in both animal and human studies and target tissue response. Interestingly, our findings in the stomach are in line with previous findings demonstrating estradiol increasing GCH-1 and DHFR in pulmonary arterial endothelial cells and MCF-7 cells, respectively [66,67]. Perhaps, improved BH_4_ synthesis levels, oxidative stress, and pro-inflammatory markers contribute to the improved nNOS expression and functionality exhibited in E_2_ treated OVX+HFD mice.

Furthermore, there is now extensive evidence supporting the role of the p38 mitogen-activated protein kinase (MAPK) pathway in regulating inflammatory responses [16,17]. p38/MAPK is a member of the large family of serine/threonine protein kinase. Many extracellular stimulants, such as hyperglycemia, can lead to activation of p38 MAPK signaling cascade. Upon activation, p38/MAPK translocates to the nucleus to mediate activity of several transcription factors to regulate gene expression and production of inflammatory mediators [16,17,18]. MAPK has been shown to be enhanced after at least 8 weeks of HFD feeding. Here, we noticed elevated levels after 12 weeks of HFD consumption. Furthermore, overexpression of GCH-1 has been shown to suppress MAPK1/2 while increasing p-MAPK expression in diabetic cardiomyocytes after superoxide production, further suggesting crosstalk between NO, nNOS, and MAPK [68]. In addition, P_4_ has been hypothesized to influence downstream activation of MAPKs [27]. Though interactions may exist in these pathways, further studies are needed to delineate distinct signaling targets between E_2_, P_4_, and their gastric receptors.

Our studies corroborate evidence by others suggesting pro-inflammatory mediators such as TNF-α and interleukin-6 (IL-6) are upregulated in diabetic patients and animal models [69]. Most importantly, previous studies suggest that several pro-inflammatory mediators have been demonstrated to be involved in gastrointestinal dysmotility by their inhibitory effects on neuronal cells as well as by direct alteration of gastric smooth muscle cell signaling via MAPK [69,70]. HFD-fed OVX mice supplemented with sex hormones, E_2_ and P_4_, displayed enhanced MAPK expression. Furthermore, studies demonstrate that blockade of p38 MAPK pathway rescues delayed GE in diabetic rats, at least in part, by inhibiting the expression of TNF-α and IL-1β [70]. In this study, we showed that TNF-α and IL-6 protein levels were significantly increased in OVX+HFD groups, in agreement with the up-regulation of inflammatory mediators reported by previous studies [33,42]. Moreover, Grover and colleagues assert that transcriptomic signatures reveal immune dysregulation in human diabetic and idiopathic gastroparesis, suggesting that cytokines may underlie abnormal GE rates in different models [47]. Interestingly, TNF-α and estradiol balance plays an important role in development of insulin resistance [71,72]. Moreover, some studies report estradiol levels inversely correlated with TNF-α levels [40], as this study showed.

Oxidative stress results from an imbalance between productions of reactive oxygen species (ROS) and antioxidant defenses1 such as catalase, superoxide dismutase (SOD), and heme oxygenase (HO-1). Oxidative stress is a well-known precipitating factor in the pathogenesis of gastroparesis [13,14]. Nuclear factor-erythroid 2 p45-related factor 2 (Nrf2) is an oxidative stress-sensitive transcription factor. Recent studies report that Nrf2 plays an important role in regulating cellular defense against oxidative stress by activating the expression of an array of antioxidant response element-dependent genes [73]. Moreover, Nrf2 has proven effective in alleviating clinical manifestations of DM in addition to regulating gastric nNOS function [40,50]. We recently demonstrated that Nrf2 activation is protective against inflammation, delayed GE, impaired NANC relaxation, and elevated serum E_2_ [38]. We sought to understand whether this mechanism was apparent in OVX mice consuming an HFD and whether E_2_ and P_4_ improve Nrf2 and Phase II enzyme protein induction. Our results demonstrated differential expression of Nrf2 in ovary intact and OVX mice consuming both ND and HFD. Furthermore, E_2_ suppressed Nrf2 expression, whereas P_4_ maintained elevated Nrf2 expression in OVX+HFD mice. Since Nrf2 levels are upregulated in response to highly oxidative stressful situations, such as obesity and hyperglycemia, our findings may suggest the E_2_ is beneficial in reducing the oxidative stress insult of HFD consumption in OVX mice. Our data further showed that E_2_ supplementation was beneficial in restoring HO-1, SOD, and catalase despite decreased Nrf2 expression. Several studies report the confounding effects of estrogens on Nrf2 in various cell types [74,75]. Recent studies from our laboratory demonstrated that activation of Nrf2 with cinnamaldehyde rescued ER levels in HFD-fed diabetic mice [38]. Moreover, E_2_ and selective ER agonists, 4, 4′, 4″-(4-Propyl-[1H]-pyrazole-1,3,5-triyl)trisphenol (PPT: ERα-selective agonist) and diarylproprionitrile (DPN: ERβ-selective agonist) improved gastric Nrf2 expression in-vitro exposed to hyperglycemic conditions [76]. Furthermore, unpublished data from our group indicated that E_2_ was beneficial in restoring gastric Nrf2 and GE in streptozotocin-induced type 1 (T1DM) diabetic mice. In contrast to in-vitro or in-vivo type 1 diabetic conditions, we observed that E_2_ downregulated gastric Nrf2 expression in HFD-induced type 2 diabetes. We speculate that, depending on the type of diabetes (type 1 vs. type 2), the underlying mechanisms of sex hormone regulation on Nrf2 are complex and require further investigation.

## 5. Conclusions

Accelerated GE (gastroparesis-like symptoms in humans) has received little attention as a complication of Gp, while most studies focus on delayed GE (gastroparesis). Our data postulate that supplementation of E_2_ and P_4_ led to improved glucose homeostasis and NO mediated gastric function in HFD induced diabetes. E_2_ and P_4_ have myriad cellular pathways to mediate stomach function. Identification of the underlying mechanistic networks is important to both public health and therapeutic issues. A better understanding of the pathogenesis of GI dysfunction in human and mouse models of obesity and diabetes will aid in developing of alternative therapeutic options. We conclude that activation of gastric ERs with E_2_ and/or selective ER modulators are a promising frontier for managing the variations in gastric motility observed in obesity-induced diabetes in addition to inflammation and oxidative stress.

## Figures and Tables

**Figure 1 antioxidants-09-00582-f001:**
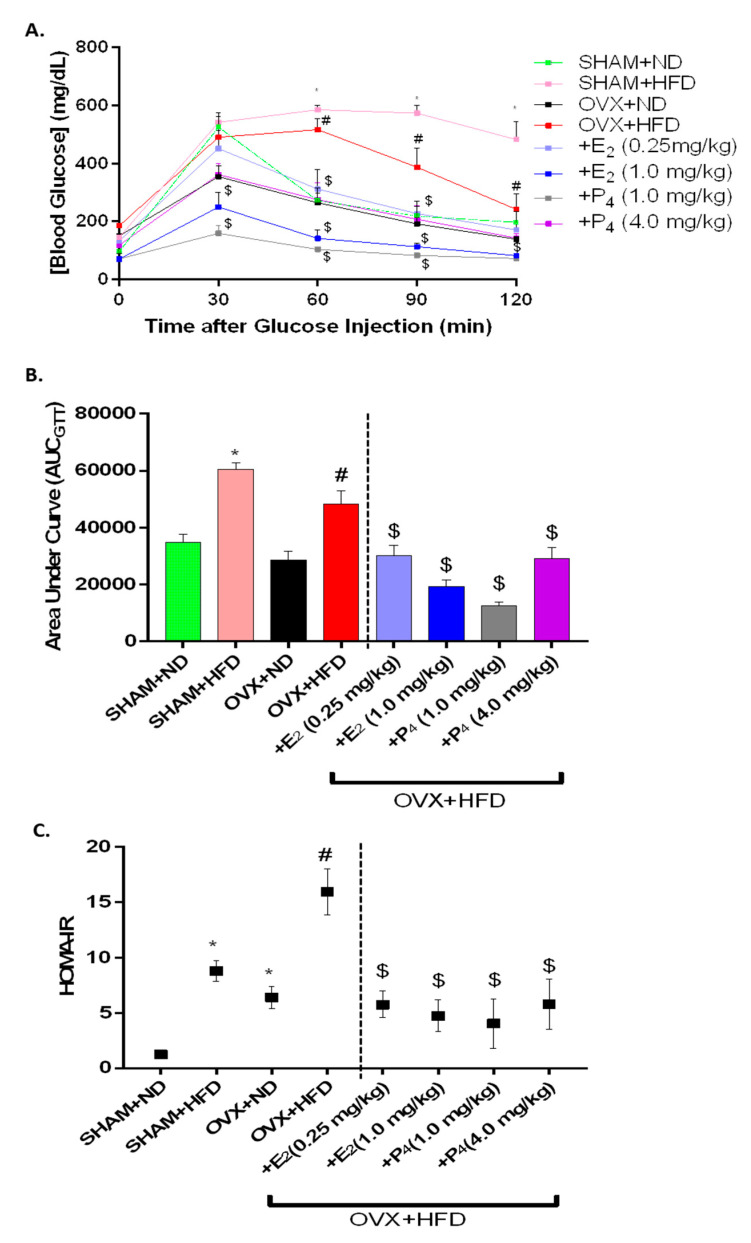
Effect of ovariectomy (OVX), 17β-Estradiol (E_2_) and Progesterone (P_4_) on glucose tolerance and insulin sensitivity in high-fat diet (HFD) fed mice. (**A**,**B**) Intraperitoneal glucose tolerance test (IPGTT) and area under curve (AUC_IPGTT)_; (**C**) homeostatic model of insulin resistance (HOMA-IR) plot was calculated using the following formula: fasting blood glucose (mg/dL) × fasting plasma insulin (μU/mL)/405. Data were analyzed using one-way ANOVA by using graph pad prism software. Values are means ± S.E.M. for *n* = 6 mice. * *p* < 0.05, significantly different from normal diet-fed sham-operated (SHAM+ND) mice. # *p* < 0.05, significantly different from normal diet-fed ovariectomized (OVX+ND) mice. $ *p* < 0.05, significantly different from high-fat diet-fed OVX (OVX+HFD) mice.

**Figure 2 antioxidants-09-00582-f002:**
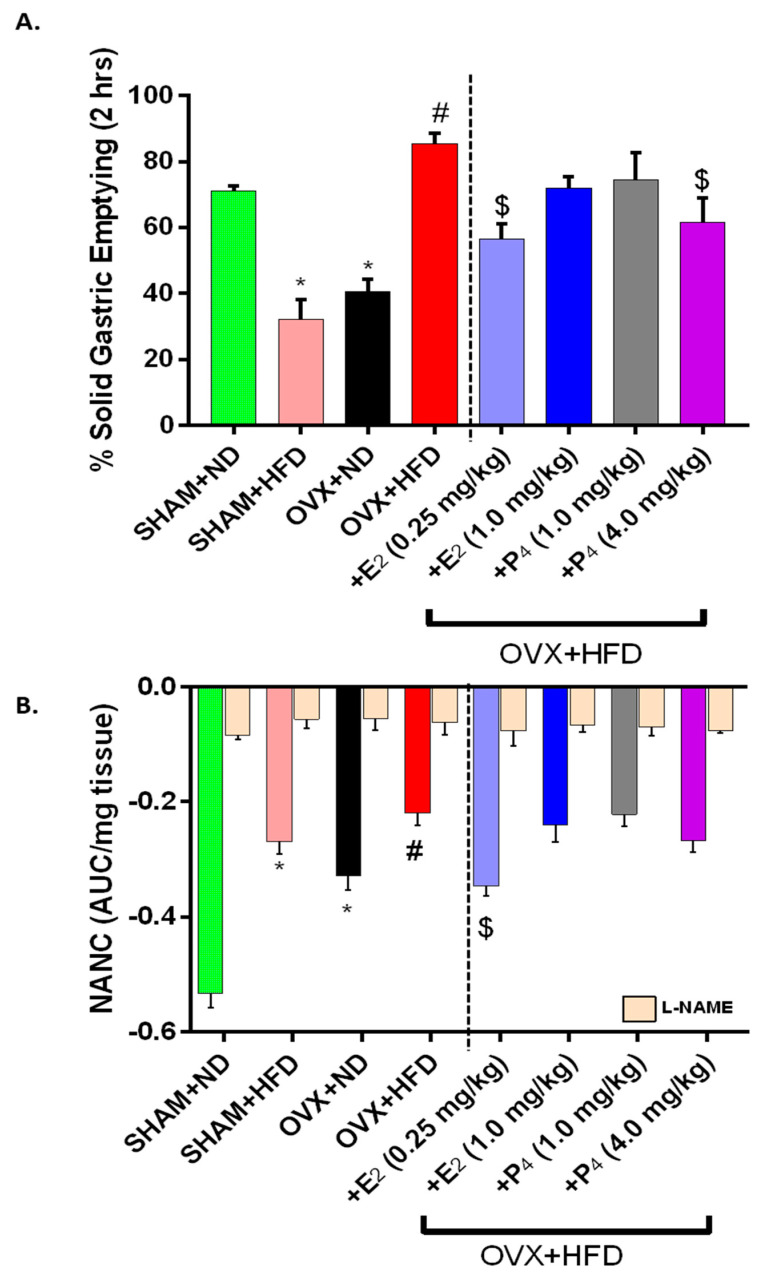
Effect of ovariectomy (OVX), 17β-Estradiol (E_2_), and Progesterone (P_4_) on (**A**,**B**) 2 h solid gastric emptying and gastric antrum nitrergic relaxation in HFD fed mice. Full and empty stomach weights were recorded; the difference estimated the remaining gastric content (GC) after 2 h of fasting. The rate of gastric emptying (GE) was measured with the equation of 1- [(Gastric Content /Food Intake) × 100. Nitrergic relaxation was measured in gastric antrum circular muscle strips at 2 Hz in an organ bath under physiological conditions. The nitric oxide (NO) dependence of the non-adrenergic non-cholinergic relaxation (NANC) relaxations was confirmed by preincubation (30 min) with the NO inhibitor nitro-L-arginine methyl ester (L-NAME; 100 μM). Area under the curve for is presented. Data were analyzed using one-way ANOVA by using graph pad prism software. Data are means ± SEM (*n* = 6). * *p* < 0.05, significantly different from normal diet-fed sham-operated (SHAM+ND) mice. # *p* < 0.05, significantly different from normal diet-fed ovariectomized (OVX+ND) mice. $ *p* < 0.05, significantly different from high-fat diet-fed OVX (OVX+HFD) mice.

**Figure 3 antioxidants-09-00582-f003:**
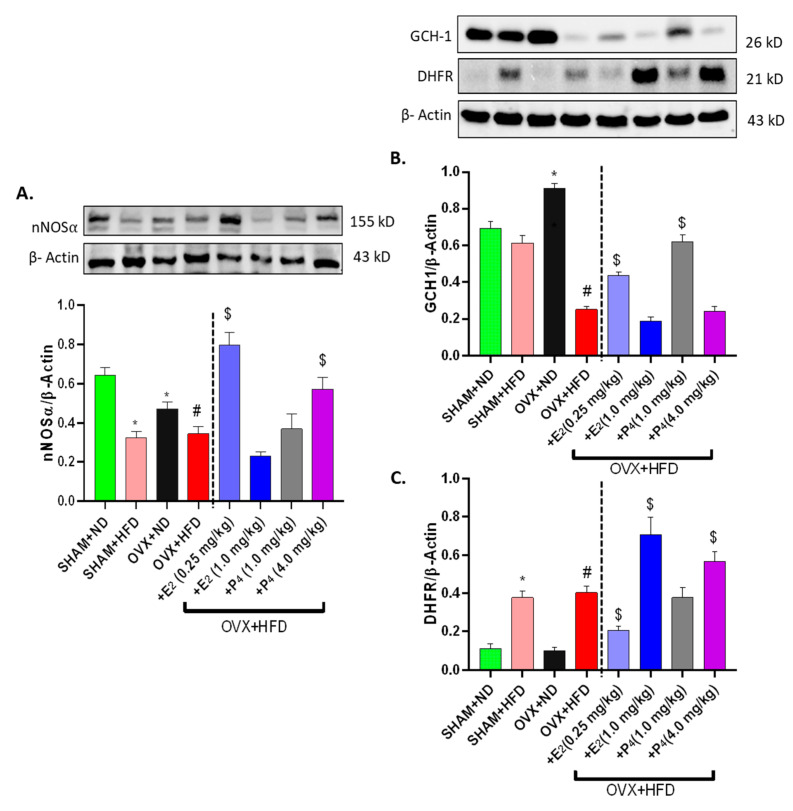
Effect of 17β-Estradiol (E_2_) and Progesterone (P_4_) on gastric neuronal nitric oxide synthase (nNOSα), GCH-1, and DHFR expressions in HFD-fed OVX mice. Representative immunoblot and densitometry analysis data for (**A**) nNOS α, (**B**) GCH-1, and (**C**) DHFR protein expression. Stripped blots were re-probed with β-actin. Data were normalized with housekeeping protein (β-actin). Bar graphs showed a ratio of target gene or protein with β-actin. Data were analyzed using one-way ANOVA by using graph pad prism software. Data are means ± SEM (*n* = 6). * *p* < 0.05, significantly different from normal diet-fed sham-operated (SHAM+ND) mice. # *p* < 0.05, significantly different from normal diet-fed ovariectomized (OVX+ND) mice. $ *p* < 0.05, significantly different from high-fat diet-fed OVX (OVX+HFD) mice.

**Figure 4 antioxidants-09-00582-f004:**
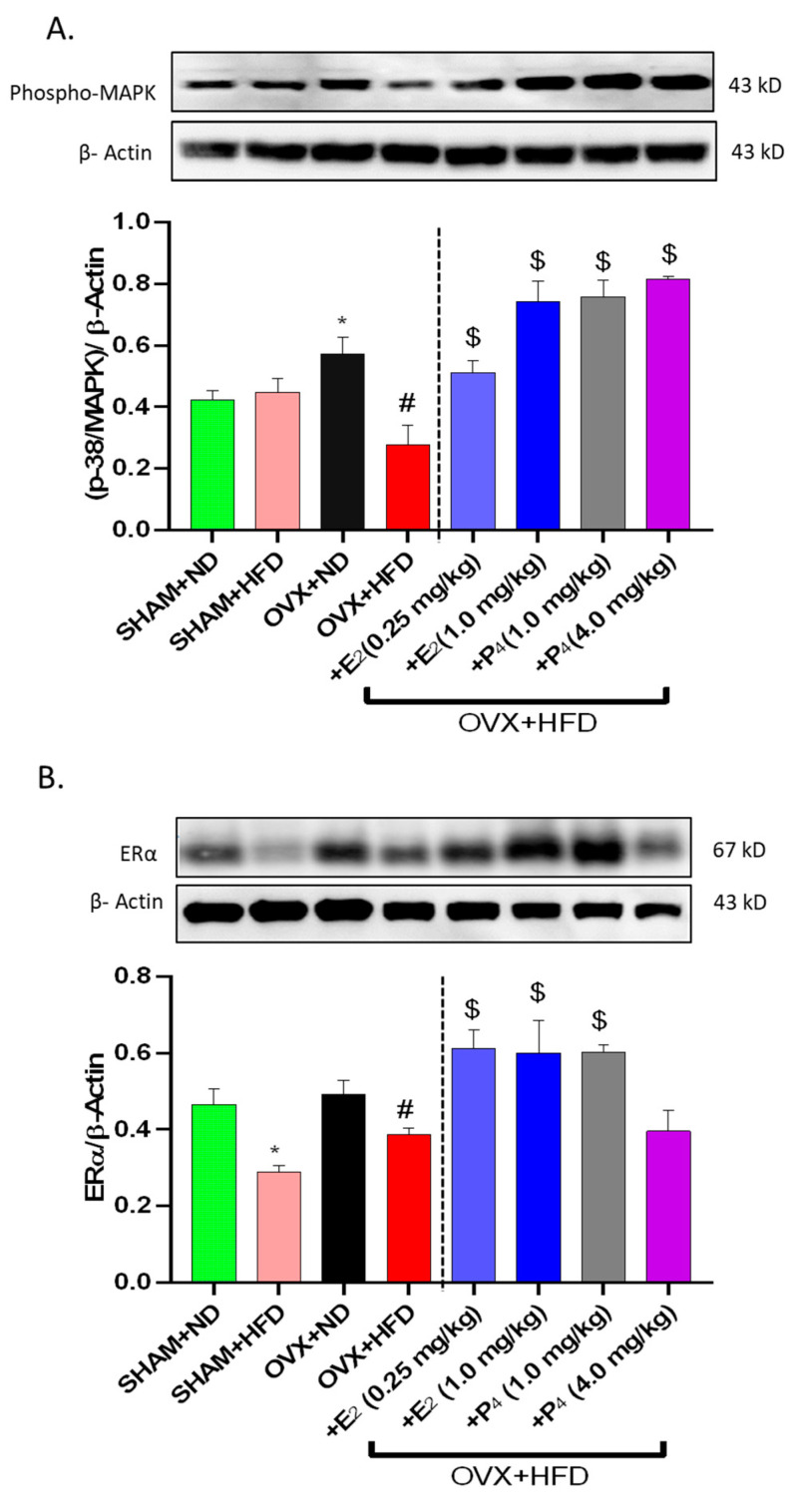
Effect of 17β-Estradiol (E_2_) and Progesterone (P_4_) on protein expressions of estrogen receptor alpha (ER α) and estrogen receptor beta (ER β) and progesterone receptor (PR) and phospho-MAPK in gastric neuromuscular tissues of OVX mice fed an HFD. Representative immunoblot and densitometric analysis data for (**A**) p-MAPK, (**B**) ER α, (**C**) ER β, (**D**) PR protein expression in female mice gastric neuromuscular tissue. Stripped blots were re-probed with β-actin. Data were normalized with housekeeping protein (β-actin). Bar graphs showed a ratio of target gene or protein with β-actin. Data were analyzed using one-way ANOVA by using graph pad prism software. The values are mean ± SEM (*n* = 4). * *p* < 0.05, significantly different from normal diet-fed sham-operated (SHAM+ND) mice. # *p* < 0.05, significantly different from normal diet-fed ovariectomized (OVX+ND) mice. $ *p* < 0.05, significantly different from high-fat diet-fed OVX (OVX+HFD) mice.

**Figure 5 antioxidants-09-00582-f005:**
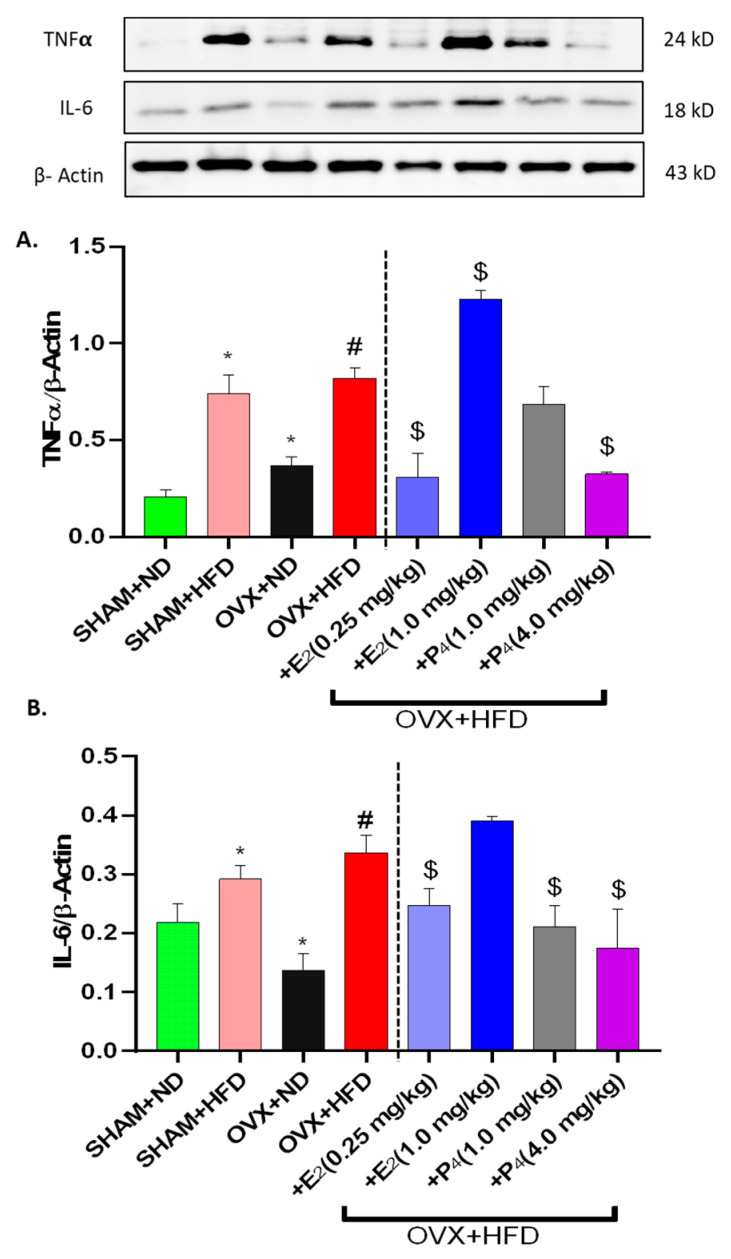
Effect of 17β-Estradiol (E_2_) and Progesterone (P_4_) on pro-inflammatory markers, interleukin-6 (IL-6) and tumor necrosis factor-α (TNFα) protein expression in HFD-fed OVX mice. Representative immunoblot and densitometric analysis data for (**A**) TNF α, (**B**) IL-6 in female mice gastric neuromuscular tissues. Stripped blots were re-probed with β-actin. Data were normalized with housekeeping protein (β-actin). Bar graphs showed a ratio of target gene or protein with β-actin. Data were analyzed using one-way ANOVA by using graph pad prism software. The values are mean ± SEM (*n* = 4). * *p* < 0.05, significantly different from normal diet-fed sham-operated (SHAM+ND) mice. # *p* < 0.05, significantly different from normal diet-fed ovariectomized (OVX+ND) mice. $ *p* < 0.05, significantly different from high-fat diet-fed OVX (OVX+HFD) mice.

**Figure 6 antioxidants-09-00582-f006:**
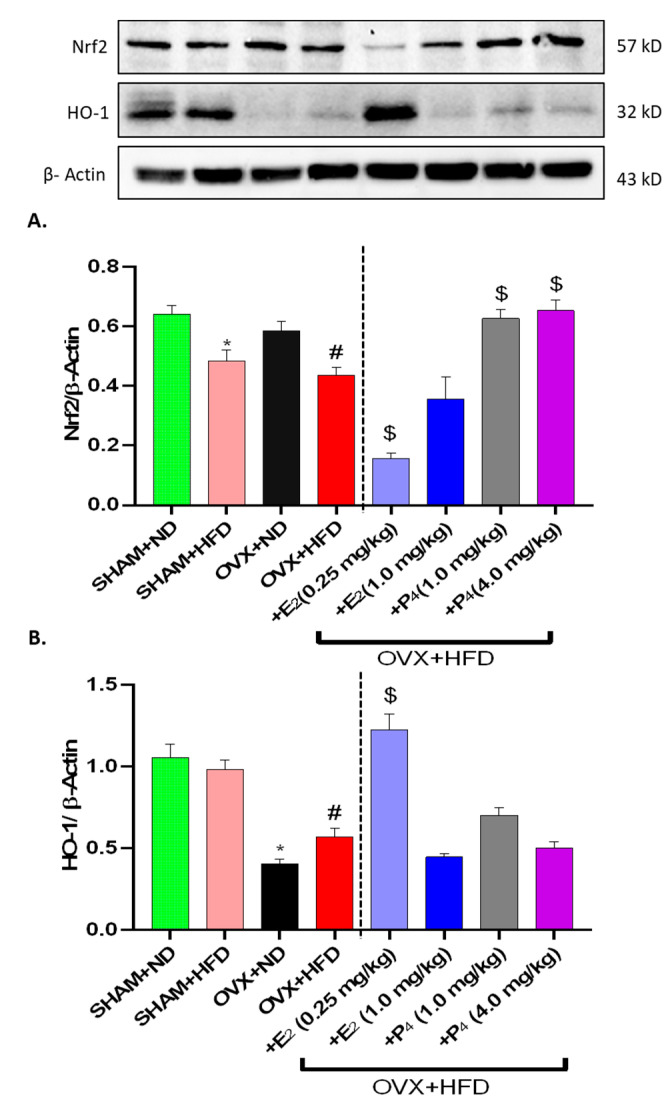
Effect of 17β-Estradiol (E_2_) and Progesterone (P_4_) on oxidative stress-responsive enzymes Nrf2, and its Phase II enzyme induction in HFD-fed OVX mice. Representative immunoblot and densitometric analysis data for (**A**) Nrf2, (**B**) heme oxygenase 1 (HO-1), (**C**) catalase, and (**D**) superoxide dismutase (SOD) in female mice gastric neuromuscular tissues. Stripped blots were re-probed with β-actin. Data were normalized with housekeeping protein (β-actin). Bar graphs showed a ratio of target gene or protein with β-actin. Data were analyzed using one-way ANOVA by using graph pad prism software. The values are mean ± SEM (*n* = 4). * *p* < 0.05, significantly different from normal diet-fed sham-operated (SHAM+ND) mice. # *p* < 0.05, significantly different from normal diet-fed ovariectomized (OVX+ND) mice. $ *p* < 0.05, significantly different from high-fat diet-fed OVX (OVX+HFD) mice.

**Figure 7 antioxidants-09-00582-f007:**
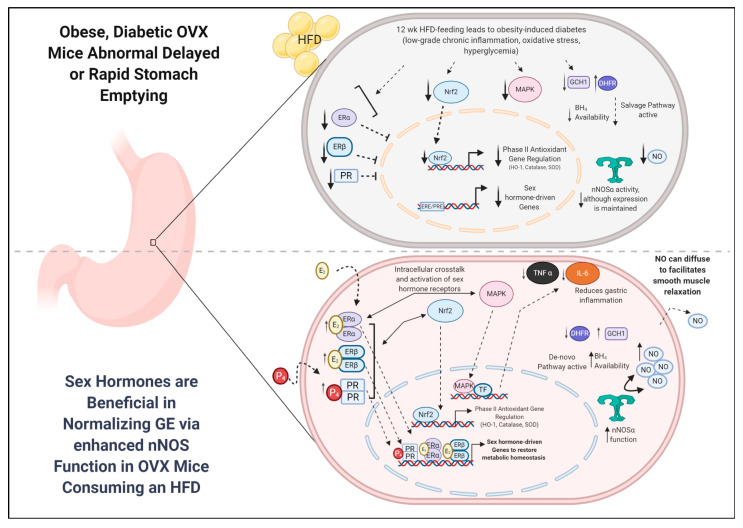
Schematic illustration depicting the hypothesized effects of 17β-Estradiol (E_2_) and Progesterone (P_4_) on mechanistic signaling of Sex hormone receptors on Nrf2, MAPK, and nNOS mediated gastric motility and gastric emptying in obesity/T2D female mice. We speculate that the sex hormone-depleted (OVX) gastric milieu may exhibit compensatory mechanisms to facilitate faster GI motility when consuming a daily HFD. Perhaps a gastrointestinal (GI) system devoid of ovarian sex hormones (chronic OVX) displays gradual changes in ER, Nrf2, and MAPK signaling in the gastric smooth muscle and enteric neurons to possibly “rewire” the female gut. The supplementation of sex hormones, E_2_ and P_4_, restored not only systemic and gastric fitness through restoration of proinflammatory pathways but also nNOS and its essential BH4 cofactor expression to normalize GE rates in obese, diabetic mice consuming a daily HFD.

**Table 1 antioxidants-09-00582-t001:** Effect of E_2_ and P_4_ on animal body and visceral adipose weights, fasting glucose, and insulin levels.

Group	*n*	Δ BW (g)	Visceral Adipose Weight (g)	Fasting Blood Glucose (mg/dL)	Fasting Insulin (ng/mL)
SHAM+ND	6	3.39 ± 1.01	0.33 ± 0.05	97.75 ± 4.29	0.24 ± 0.05
SHAM+HFD	6	19.25 ± 2.61 *	2.37 ± 0.24 *	208.00 ± 11.16 *	0.98 ± 0.19 *
OVX+ND	6	4.39 ± 1.25	0.78 ± 0.18 *	150.60 ± 6.28 *	0.81 ± 0.09 *
OVX+HFD	6	13.82 ± 0.68 ^#^	3.35 ± 0.22 ^#^	185.33 ± 8.25 ^#^	1.09 ± 0.22 ^#^
+E_2_ (0.25 mg/kg)	6	15.15 ± 1.05	2.40 ± 0.37 ^$^	151.00 ± 3.13 ^$^	0.903 ± 0.14
+E_2_ (1.0 mg/kg)	6	5.78 ± 1.25 ^$^	0.17 ± 0.04 ^$^	108.33 ± 6.62 ^$^	0.867 ± 0.20
+P_4_ (1.0 mg/kg)	6	10.07 ± 0.75 ^$^	0.68 ± 0.11 ^$^	70.75 ± 3.42 ^$^	1.03 ± 0.46
+P_4_ (4.0 mg/kg)	6	8.07 ± 1.36 ^$^	0.86 ± 0.14 ^$^	118.25 ± 5.41 ^$^	1.03 ± 0.31

* Denotes *p* < 0.05 compared to (SHAM+ND) group; # Denotes *p* < 0.05 compared to (OVX+ND) group; $ Denotes *p* < 0.05 compared to (OVX+HFD) group.

**Table 2 antioxidants-09-00582-t002:** Concentrations of serum 17β-Estradiol (E_2_), Progesterone (P_4_), total nitrite, and adiponectin in experimental groups.

	Serum 17β-Estradiol (ng/L)	Serum Total Nitrite (μM)	Serum Progesterone (ng/mL)	Serum Adiponectin (μg/mL)
SHAM+ND	34.72 ± 4.22	19.87 ± 0.39	5.69 ± 0.66	16.56 ± 2.19
SHAM+HFD	44.49 ± 3.99 *	16.05 ± 0.55 *	8.65 ± 1.04 *	6.29 ± 0.12 *
OVX+ND	21.80 ± 3.42 *	15.76 ± 0.64 *	3.46 ± 1.38	11.46 ± 0.54 *
OVX+HFD	39.07 ± 5.05 ^#^	19.42 ± 0.48 ^#^	6.41 ± 1.23 ^#^	5.01 ± 0.16 ^#^
+E_2_ (0.25 mg/kg)	38.10 ± 1.69	21.19 ± 0.64 ^$^	2.60 ± 0.59 ^$^	9.88 ± 0.23 ^$^
+E_2_ (1.00 mg/kg)	44.63 ± 4.27	16.66 ± 0.19 ^$^	3.19 ± 1.34 ^$^	10.14 ± 0.56 ^$^
+P_4_ (1.00 mg/kg)	38.41 ± 3.64	16.94 ± 0.69 ^$^	7.78 ± 1.45	6.74 ± 0.71
+P_4_ (4.00 mg/kg)	23.67 ± 6.29 ^$^	18.87 ± 0.21	9.44 ± 0.93	9.25 ± 0.57 ^$^

* Denotes *p* < 0.05 compared to (SHAM+ND) group; # Denotes *p* < 0.05 compared to (OVX+ND) group; $ Denotes *p* < 0.05 compared to (OVX+HFD) group.

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
