# Peer review of "Supplementation of 17β-Estradiol Normalizes Rapid Gastric Emptying by Restoring Impaired Nrf2 and nNOS Function in Obesity-Induced Diabetic Ovariectomized Mice"

_antioxidants, 2020, doi:10.3390/antiox9070582_

Round 1
Reviewer 1 Report
The present manuscript is an original article which reports the impact of 17β-estradiol on gastric emptying in obesity-induced diabetic ovariectomized mice. This manuscript provides interesting information although some crucial aspects are not clearly demonstrated or need to be explained.
However, some points described below should be included in order to improve this manuscript:
- Description “sex hormone” should be replaced by “17β-estradiol and progesterone” in the whole manuscript.
- The authors stated that loss of enteric neuronal nitric oxide synthase (nNOS) and nitric oxide (NO)-mediated relaxation of gastric smooth muscle as the primary culprits of Gp in diabetic patients. Why the authors did not analyze the nNOS+ enteric neuronal population using immunohistochemistry or immunofluorescence technique?
- Table 1 legend has to be improved; body weight or visceral adipose weight is not metabolic parameters.
- Line 179: Statistical analysis. Data were presented as the mean and not only SE but also SEM, like in the case of all figures.
- Line 199 – 200: according to Table 1 lower level of glucose but not insulin was identified in E2 and P4-treated mice vs. HFD-fed OVX mice.
- Table 2: The level of 17β-estradiol in OVX mice is high, why?
- Why only nuclear estrogen receptors are analyzed. Membrane-bound estrogen receptor, i.e. GPER which action is related to G-protein and affects MAPK signaling should be also taken into consideration.
Author Response
Response: We thank this reviewer for providing insightful critiques. We carefully reviewed the comments and revised the manuscript as suggested by this reviewer.
Please see our point-by-point responses below.
- Description “sex hormone” should be replaced by “17β-estradiol and progesterone” in the whole manuscript.
Response: Thank you for bringing this to our attention. We agree that in many instances, “sex hormones” is not descriptive and we have therefore replaced sex hormones with 17β-estradiol (E2) and progesterone (P4) where appropriate in the revised manuscript. There were many occurrences as outlined in the tracking changes, but include Lines: 39, 43, 83, 94, 123, 130, 134, 144, 193, 225, 241, 245, 278, 281, 318, 321, 326, 334, 341, 343, 356, 376, 393, 395, 402, 435, 474, 488, 494, 543, 565, 571, 887, 957, 969, 983, 993, 1005, 1016, 1029.
- The authors stated that loss of enteric neuronal nitric oxide synthase (nNOS) and nitric oxide (NO)-mediated relaxation of gastric smooth muscle as the primary culprits of Gp in diabetic patients. Why the authors did not analyze the nNOS+ enteric neuronal population using immunohistochemistry or immunofluorescence technique?
Response: We cannot deny that uncovering the changes in nNOS density and distribution through immunohistochemical techniques would bolster the strength of our assertions. However, our laboratory has an extensive experience in many of the principles of nNOS-mediated gastric motility, particularly in female diabetes. In this study, we demonstrate diminished nNOS protein expression, nitrergic relaxation (as confirmed by using NOS inhibitor L-NAME, and total nitrite production in neuromuscular antrum specimens suggesting and reinforcing the critical role of nNOS in mediating gastric motility. Since, the above data are complementing each other; we did not attempt to analyze the nNOS+ enteric neuronal population using IHC/IFC methods. However, it is our goal to submit a separate manuscript mapping the gradual changes in nNOS, ERs and Nrf2 in addition to colocalization of these markers in female gastric smooth muscle at various time points of all the groups. We feel that this data would be best suited for complete presentation in a future manuscript. We have incorporated this rationale in the discussion of the revised manuscript in Lines: 444-448
- Table 1 legend has to be improved; body weight or visceral adipose weight is not metabolic parameters.
Response: We agree with the comment and have rephrased the legend to be more descriptive:
The revised legend is “Effect of E2 and P4 on Animal Body & Visceral Adipose Weights, Fasting Glucose and Insulin Levels. (Revised in Lines: 877-878).
Also corrected in the Results Heading: Supplementation of E2 and P4 Prevent Body Weight Gain, Adiposity and Hyperglycemia in Obese, Diabetic OVX Mice (Line: 225)
- Line 179: Statistical analysis. Data were presented as the mean and not only SE but also SEM, like in the case of all figures.
Response: This typo was corrected in the revised manuscript.” Data were presented as the mean ± standard error of the mean (SEM). “ (corrected in Line 220)
- Line 199 – 200: according to Table 1 lower level of glucose but not insulin was identified in E2 and P4-treated mice vs. HFD-fed OVX mice.
Response: We understand that this was not fully explained in the results section. This data is now clarified in the revised results and discussion sections (Lines: 241-244).
” As reported earlier, our current data reinstate that in-vivo supplementation of E2 and P4 markedly attenuate elevated fasting glucose levels , visceral adipose weight and weight gain in HFD-fed OVX mice[18,42,43]. In contrast, as shown in Table 1 that fasting insulin levels were unchanged in mice receiving E2 and P4.”
- Table 2: The level of 17β-estradiol in OVX mice is high, why?
Response: Thank you for bringing this to our attention and we understand your concern. Our serum E2 levels in OVX group are comparable with previous reports in the literature employing similar animal models (please see below references 1 - 5). Our results in Table 2 further corroborate the assertion that E2 levels are significantly lower in OVX (OVX+ND) compared to ovary-intact (SHAM+ND) group.
References:
- Fejes-Szabó, A., Spekker, E., Tar, L., Nagy-Grócz, G., Bohár, Z., Laborc, K. F., Vécsei, L., & Párdutz, Á. (2018). Chronic 17β-estradiol pretreatment has pronociceptive effect on behavioral and morphological changes induced by orofacial formalin in ovariectomized rats. Journal of pain research, 11, 2011–2021. https://doi.org/10.2147/JPR.S165969
- Haisenleder, D. J., Schoenfelder, A. H., Marcinko, E. S., Geddis, L. M., & Marshall, J. C. (2011). Estimation of estradiol in mouse serum samples: evaluation of commercial estradiol immunoassays. Endocrinology, 152(11), 4443–4447. https://doi.org/10.1210/en.2011-1501
- Song CH, Kim N, Lee SM, et al. Effects of 17β-estradiol on colorectal cancer development after azoxymethane/dextran sulfate sodium treatment of ovariectomized mice [published correction appears in Biochem Pharmacol. 2019 Oct;168:339-340]. Biochem Pharmacol. 2019;164:139-151. doi: 1016/j.bcp.2019.04.011
- Kaliannan, K., Robertson, R.C., Murphy, K. et al.Estrogen-mediated gut microbiome alterations influence sexual dimorphism in metabolic syndrome in mice. Microbiome 6, 205 (2018). https://doi.org/10.1186/s40168-018-0587-0
- Michael E. Wyde, John Seely, George W. Lucier, Nigel J. Walker, Toxicity of Chronic Exposure to 2,3,7,8-Tetrachlorodibenzo-p-dioxin in Diethylnitrosamine-Initiated Ovariectomized Rats Implanted with Subcutaneous 17 β-Estradiol Pellets, Toxicological Sciences, Volume 54, Issue 2, April 2000, Pages 493-499, https://doi.org/10.1093/toxsci/54.2.493
- Why only nuclear estrogen receptors are analyzed. Membrane-bound estrogen receptor, i.e. GPER which action is related to G-protein and affects MAPK signaling should be also taken into consideration.
Response: Thank you for this comment. We agree that ER signaling mechanisms are diverse and include membrane-bound estrogen receptors to facilitate many intracellular actions within the various cell types. Particularly, the non-genomic activity of ERs is mediated by both cytosolic (ER ⍺, ?) and membrane-bound (GRP30/GPER, and post-translationally modified m-ER⍺ /m-ER?). We further agree that GPR30 mediates many of the non-genomic signaling cascades including MAPK and AKT that occur within minutes to hours of activation that may ultimately influence gene regulation. However, since our study primarily focused on the chronic effects of OVX (12 weeks following OVX), in addition to daily HFD consumption and sex hormone supplementation, our primary focus was to elucidate the genomic changes that occur over days to weeks in the development of obesity and diabetes, as opposed to the quick, often transient non-genomic actions. Furthermore, cytosolic ERs are widely known to interact with MAPK. In this model, we expect the genomic changes in our target proteins result from the actions of cytosolic and nuclear ERs, therefore we have not explored whether non-genomic signaling is altered in this study. GPR30 and ER beta have also been shown to display a similar distribution in the GI-system and have been implicated in mediating the gastric motility and visceral sensations in ex-vivo preparations. Moreover, we plan to consider the role of selective activation of GPERs and non-genomic mechanisms in regulating GE when preparing our future studies. We have included this information in the discussion of the revised manuscript (Lines: 458-468)
Reviewer 2 Report
This manuscript is well written and possesses an easy to understand flow to the information provided. The data obtained, and the subsequent discussion are fine. The Introduction paragraphs do well to transition between topics, with great final sentences that summarize the point being made in the denser background material. The introduction highlights variables that are targeted in the experiments, like BH4, GCH1, and DHFR. Quick factual statements were made and discussed in greater detail later. This allows for the natural expansion of information and made it easy to follow complex topics. The Discussion section is also very good. It summarizes the information into impactful interpretations. The proposed mechanism of effect present in Figure 7 is a great supplemental aide for the points identified in the discussion.
To edit, here some suggestions:
Line 119: ‘…previous day was and discarded.’
For all figures referring to experimental groups:
Consider changed the +HFD group title to OVX+HFD. This would help to keep the titles consistent between the sham groups and ovx groups.
Paragraph lines 294 to 302:
I think the Figures should be referring to 3b and 3c, rather than 3c and 3d? (also present in the Figure 3 title)
Line 319: I think Figure 6A should read Figure 4 b,c?
Lines 366 and 368: Should the Figure references include Fig 6 b,-d? and not just Fig6 b,c?
Line 538: I think you can delete the ‘in OVX-HFD groups as this was stated earlier in the sentence.
Author Response
Response: Thank you for the review and constructive critique of our manuscript. We also thank this reviewer for the comment that our manuscript is well written.
We hope to address your concerns in the point-by-point responses below.
To edit, here some suggestions:
- Line 119: ‘…previous day was and’
Response: This typo is revised in Line 155 of the revised manuscript, and written as:
” Once feeding began, fresh chow was provided daily and any remaining chow from the previous day was discarded”
- For all figures referring to experimental groups: Consider changed the +HFD group title to OVX+HFD. This would help to keep the titles consistent between the sham groups and ovx groups.
Response: We agree that this modification would enhance the manuscript with consistent labels. We changed “+HFD” in the figures to “OVX+HFD,” accordingly.
- Paragraph lines 315 to 318: I think the Figures should be referring to 3b and 3c, rather than 3c and 3d? (also present in the Figure 3 title)
Response: Thank you for bringing this typo to our attention. This revised paragraph in Lines 353-356 and is presented:
“Although we observed no change in GCH-1 expression in HFD-fed Sham-operated mice, GCH-1 protein expression was elevated in OVX+ND mice and was significantly (p<0.05) repressed in HFD-fed OVX mice (Figure 3b). Both E2 and P4 significantly (p<0.05) increased GCH-1 expression in OVX-HFD mice. Interestingly, we found DHFR expression was significantly upregulated in instances which GCH-1 expression was diminished, in HFD-fed groups (p<0.05) (Figure 3c). In addition, E2 and P4 had dose-differential effects on GCH-1 and DHFR expression, suggesting hormonal regulation of BH4 synthesis enzymes.”
We also revised figure number in the legend section (Line:985)
- Line 319: I think Figure 6A should read Figure 4 b,c?
Response: Yes, the typo was revised in line 326-332:
“Moreover, representative western blots of ERα and ERβ levels in OVX mice following HFD feeding are presented in Fig. 4 b, c. Our data indicate that HFD feeding is associated with a depletion in sex hormone receptors, as compared to ND-fed mice.”
We also revised figure number in the legend section (Line: 997)
- Lines 366 and 368: Should the Figure references include Fig 6 b,-d? and not just Fig6 b,c?
Response: Yes, the typo was revised in line 356-358:
“Catalase, SOD, and HO-1 were maintained in HFD-fed Sham mice compared to ND-fed counterparts (Fig 6 B-D). OVX led to an upregulation of SOD1, although decreased expression of cellular Catalase and HO-1 was observed (Fig 6 B-D) . E2 and P4 administration differentially regulated Phase II antioxidant enzyme expression; both increased expression of SOD, HO-1 and Catalase at different concentrations (Fig 6 B-D).”
We also revised figure number in the legend section (Lines # 1018-1019)
- Line 538: I think you can delete the ‘in OVX-HFD groups as this was stated earlier in the sentence.
Response: We agree; this is a redundant typo, and has been revised in Line 524:
“In this study, we showed that TNF-α and IL-6 protein levels were significantly increased in OVX-HFD groups, in agreement with the up-regulation of inflammatory mediators reported by previous studies”
Round 2
Reviewer 1 Report
The authors significantly improved the manuscript.